# Automatic Detection of Oral Squamous Cell Carcinoma from Histopathological Images of Oral Mucosa Using Deep Convolutional Neural Network

**DOI:** 10.3390/ijerph20032131

**Published:** 2023-01-24

**Authors:** Madhusmita Das, Rasmita Dash, Sambit Kumar Mishra

**Affiliations:** 1Department of Computer Application, Siksha ‘O’ Anusandhan Deemed to be University, Bhubaneswar 751030, India; 2Department of Computer Science and Engineering, Siksha ‘O’ Anusandhan Deemed to be University, Bhubaneswar 751030, India; 3Department of Computer Science and Engineering, SRM University-AP, Guntur 522240, India

**Keywords:** deep convolutional neural network, histopathological images, oral cancer

## Abstract

Worldwide, oral cancer is the sixth most common type of cancer. India is in 2nd position, with the highest number of oral cancer patients. To the population of oral cancer patients, India contributes to almost one-third of the total count. Among several types of oral cancer, the most common and dominant one is oral squamous cell carcinoma (OSCC). The major reason for oral cancer is tobacco consumption, excessive alcohol consumption, unhygienic mouth condition, betel quid eating, viral infection (namely human papillomavirus), etc. The early detection of oral cancer type OSCC, in its preliminary stage, gives more chances for better treatment and proper therapy. In this paper, author proposes a convolutional neural network model, for the automatic and early detection of OSCC, and for experimental purposes, histopathological oral cancer images are considered. The proposed model is compared and analyzed with state-of-the-art deep learning models like VGG16, VGG19, Alexnet, ResNet50, ResNet101, Mobile Net and Inception Net. The proposed model achieved a cross-validation accuracy of 97.82%, which indicates the suitability of the proposed approach for the automatic classification of oral cancer data.

## 1. Introduction

In India, the rate of oral cancer, specifically OSCC, is increasing, and the main reason behind this rise is alcohol consumption and tobacco chewing, etc. The death rate of oral cancer is also high in India. There are commercial advertisements regarding the disadvantage of tobacco and alcohol consumption; however, due to the lack of understanding and inadequate knowledge, people are still in the habit of alcohol and tobacco, which causes an increase in the number of oral cancer patients [1,2]. One of the international agencies for research conducted a survey and predicted that the number of cancer patients will increase from 1 million in 2012 to more than 1.7 million by 2035. This implies the death rate will also increase from 680,000 to 1–2 million by 2035 [3]. Hence, it is utmost important to detect the OSCC at its early stage so that the treatment can be started as early as possible and the death rate due to oral cancer can be reduced. With the advancement of technology, continuous research has been performed by many researchers for the early detection of oral cancer and, at the same time, enormous amounts of oral cancer data are collected and made available for research purposes. One of the challenging tasks for doctors is to correctly predict the type and stage of cancer. The oldest method used by doctors is the physical screening in the first stage and, after that, for the conformity purpose, biopsy is used. With the improvement in computer technology, many machine learning techniques along with image processing techniques are used by researchers to predict the type as well as stage of cancer, and that can help the doctors to give a better treatment to the OSCC patient. Among different imaging techniques, the histopathological imaging technique is more suitable for OSCC diagnosis. Thus, in this work, the author has focused on histopathological images of OSCC for the detection and classification problem.

Certain features are seen in the cell morphology, which can also be found in digital histopathological images of OSCC. Thus, it is the fundamental task of various machine learning algorithms to extract those features that can be mapped with the original cell morphology and predict the presence or absence of cancer cells accurately. A few successful supervised and unsupervised machine learning algorithms are support vector machines, neural networks, decision tree, fuzzy, genetic algorithms, k-NN, kernel PCA, etc., used for medical histopathological image classification problems [4,5,6,7,8,9]. However, in the current era, deep learning techniques, specifically convolutional neural networks (CNN), have become more popular among researchers for the image analysis problem. It is already proven that CNN outperforms for different computer vision problems mostly for object detection; however, in recent studies, it has shown that CNN can also outperform for different low-level image processing problems, such as restoration, denoising and mitosis detection, and these are the general problem in histopathological image analysis. CNN is a special type of artificial neural network (ANN). The ANN was inspired by biological neural structures and it has a learning ability like biological neurons. Because of these characteristics, a neural network can generate new dataset from the existing ones and can discover optimal outputs. CNN can observe new samples and after generalization can produce new learning rules from the sample, and this learning rule can be used to decide the output when an unseen dataset is used. The recent challenges in the area of medical imaging are to use these CNN approaches to handle the classification problem. The efficiency of deep learning, mainly in CNN, is to find out the architecture, which meets the issues of medical image classification, so that the predictable outcome can be improved [10,11].

### 1.1. Contribution

The application of deep learning techniques is quite high in several types of cancer; however, limited research analysis has been done using histopathological OSCC images. This research deliberates the classification of oral cancer data samples using histopathological images. The classification outcome can be used as the input for other tasks like the extraction of the nucleus feature classification and predicting various stages of cancer. Our main task is to design an optimal CNN model which can achieve the automatic detection of oral cancer from the histopathological images.

### 1.2. Organization

The rest of the paper is arranged as follows: related work is discussed in Section Two. The methods and techniques used are discussed in Section Three. In Section Four, the proposed methodology along with the proposed CNN model are highlighted. Experimental studies are discussed in Section Five; performance evaluation and result analysis are completed in Section Six. Finally, in the Section Seven, the conclusion is elaborated.

## 2. Related Work

Machine learning algorithms became highly successful in disease prediction models; however, the performance of the models degrades in the case of an insufficient and imbalance dataset. Models developed using insufficient dataset face the problem of overfitting. An overfitted model’s prediction is either underestimated or overestimated, which leads to poor generalization in real time of the dataset. To address the problem of model overfitting, the author of [12] used the penalized regression method for the risk prediction model. Similarly, to deal with insufficient data, in the work [13], the author discusses a novel age adoption algorithm based on feature compensation and soft decision threshold adjustment for diabetes mellitus risk prediction. In [14], a detailed survey is conducted on a few shot learning techniques to deal with the insufficient dataset. Various image data augmentation techniques are discussed in [15] to deal with the insufficient dataset problem.

Research on the use of machine learning algorithms for the detection and classification of cancer almost covers all the varieties of cancer, which include brain cancer, skin cancer, breast cancer, lung cancer, prostate cancer, oral cancer, etc. Here, the author presents a comprehensive study on the area of oral cancer.

In the literature, for oral cancer classification and for its early detection, different machine learning and deep learning techniques are widely used. Most of the research is done on the efficiency of different machine learning algorithms for the prediction of oral sub-mucous fibrosis (OSF) [16,17,18]. The advanced stage of OSF can lead to oral cancer.

A detailed study was done on the classification of OSCC by T.Y Rahaman et al. [19,20]. Rahaman et al., used Support Vector Machine (SVM) and Linear Discriminant Classifier (LDA) for the binary classification of the OSCC images. In both SVM and LDA, they used texture, color and shape features for the classification. In 2017, they used two sets of datasets, which were collected from Ayursundra healthcare Pvt. Ltd. and Dr. Borooah Cancer Research Institute, respectively, for the experiment. In the first dataset, there were a total of 110 normal images and 113 malignant images. In the second dataset, 86 normal and 88 malignant images were there. In the research work, Rahaman et al., applied SVM on 400× magnification images and achieved acceptable accuracy [19].

In 2012, Muthu et al. [21] applied a few classifiers, namely fuzzy, decision tree, Gaussion mixture model (GMM), K-nearest neighbor and probabilistic neural network, for the classification of OSF, and they found that the fuzzy classifier gave the highest accuracy of 95.7% as compared to other classifiers.

Muthu et al. also considered morphological and texture features in another study, for the classification of OSF, and in that study they applied SVM and GMM and found out that SVM gives better results compared to the GMM model [22].

Anuradha. K. et al. reflect the use of distinctive features, such as energy, entropy, contrast, correlation and homogeneity, along with the SVM classifier for the classification of the oral tumor as malignant or benign [23].

T. Belvin et al. used 16 malignant images with 192 patches, which were collected from the Himalayan Institute of Medicine, Deradun, for the multiclass classification purpose, and they applied a backpropagation-based artificial neural network model (ANN) [24].

D. Devkumar et al., in 2015, considered 10 numbers of OSCC patients and collected 30 labeled images from Tata medical center, Kolkata. Keratinization and keratin pearls were cast off as features for the keratinization index to grade the OSCC images into three grades, namely grade I, grade II and grade III. This keratinization index is used when the image magnification is as low as (4X) [25].

In 2018, D. Devkumaret et al., collected 126 images with the size 2048×1536 from Barsat Cancer Research and Welfare Center, West Bengal, and they identified various layers, such as epithelial, subepithelial, keratin region and keratin pearls, by applying CNN with the Gober filter and Random Forest, and they achieved an accuracy of 96.88% [26].

In the above studies, the research for the classification of oral cancer using machine learning techniques is stated. From the literature, it is claimed that the prediction of oral cancer is effective for early detection and diagnosis of the disease, and it is also found that the SVM classifier gives comparatively better results than other machine learning algorithms such as decision tree, ANN, Random Forest and KNN. From the literature, it can be seen that most authors have used 10-fold cross-validation method for the evaluation process and different measures, such as accuracy, sensitivity, specificity and ROC curve, are also used. 

The selection of the most appropriate algorithm depends on many factors such as the size of the dataset, the image magnification, the noise in the image, etc. The frequently appearing issue in the above literature is the small size of the dataset. For the classification algorithm, the proper amount of training and testing the dataset is required so that the validation accuracy can achieve a promising percentage. Along with the smaller dataset, the other limitations are poor dataset quality and inappropriate feature selection. For the automatic classification of a cancer dataset, it is of the utmost important to use the best feature for the classification purpose, which is a very promising task. In this scenario, deep learning algorithms achieved superior classification accuracy without manual feature extraction requirements.

There are many papers available in the literature for oral cancer prediction, but according to our best knowledge, a limited amount of work has been done using histopathological images along with deep learning advancement. Due to the availability of high-performance GPUs, deep learning algorithms can be used for histopathological images, and in the case of deep learning algorithms, manual feature extraction of the input histopathological images is not required [27,28]. For the classification of histopathological oral cancer images, Navarun et al. [29] compared the use of transfer learning with a CNN model for the classification of histopathological images of oral cancer. The transfer learning is used with pretrained CNN models, namely Alexnet, VGG-16, VGG-19 and ResNet-50. ResNet-50 outperform the other model with an accuracy of 92.15%, whereas Navarun et al. proposed a CNN model and applied it to the same dataset and achieved an accuracy of 97.5%. In the study [29], it is concluded that the CNN model outperforms the four pretrained transfer learning models.

Santisudha et al. suggested deep learning approaches [30,31] whose accuracy is comparable with other state-of-the-art models. In this work [30], the authors suggested three variants of improved ResNet-based models. Out of three models, the best one was chosen for oral cancer image classification. In the work [31], the authors suggested a modified capsule network for the classification task of oral histopathological images and achieved an accuracy of 97.35%. 

There are many studies found in the literature that are associated with deep learning for biopsy image classification [32,33]. Thus, deep learning approaches are predicted to outperform other machine learning approaches in the classification problem, without manual feature extraction requirements. However, to our best knowledge, limited work has been done in the domain of histopathological images of oral cancer cells using deep learning approaches. The literature reveals the importance of deep learning methods for classification purposes.

Inspired by the above literature analysis, this proposed study considers the binary classification of oral histopathological images using CNN architecture of different variations and layers. The outcomes of all variations are compared, and the best-performing model for classification is determined.

## 3. Methods and Techniques Used

In traditional pattern recognition problems, relevant features are extracted by the export of the area manually, and then those features are submitted to the simple neural network for the classification task [34]. However, in deep learning, the relevant features are automatically extracted and used for the solution of the problem. Deep learning is a type of neural network, which takes input in the form of metadata and process these data through several layers to compute the output [35].

In this research work, one of the deep learning techniques, CNN, is used for the classification of oral cancer histopathological images. CNN mainly consists of six layers: (i) input layer, (ii) convolution layer, (iii) pooling layer, (iv) flattening layer, (v) fully connected layer and (vi) output layer [36,37,38].

Input Layer: In this layer, the input is given, and it is converted into a matrix of pixels, where the input is an image.

Convolution Layer: The convolution layer is used to extract features by using a weight matrix in the form of filter. In each layer of convolution, one set of feature maps is generated according to Equation (1).
(1)$$Y_{p}^{\;r}=f\left[\sum_{o\in N_{p}}\left(Y_{o}^{r-1}\times M_{op}^{r}+a_{p}^{r}\right)\right]$$

Here, $N_{p}$ represents the input image set, and $Y_{p}^{\;r}$ is the p^th^ feature set of the r^th^ layer. ∗ represents convolution function. $Y_{o}^{r-1}$ represents the o^th^ feature map of the r−1 layer. $M_{op}^{r}$ represents the filter connecting the p^th^ feature map of the r layer and the o^th^ feature map of the r−1 layer. $a_{p}^{r}$ is the bias. f( ) represents the nonlinear activation functions used such as ReLU, sigmoid and tanh.

Pooling layer: The pooling layer is used to reduce the dimension of the feature map. The redundant and unnecessary features are discarded here. The pooling layer is used in between the subsequent convolution layers [39]. In max pooling methodology, the maximum value of patch is considered as the output and a reduced feature map is generated. In the case of average pooling instead of maximum value, the average value of the pixels is considered for each block. Equation (2) is the mathematical expression for the max pooling.
(2)$$Y_{p}^{\;r}=f\left[\beta_{p}^{r}downcast\;\left(Y_{o}^{r-1}\right)+a_{p}^{r}\right]$$

Here, downcast ( ) is the subsampling function and β represents the subsampling coefficient’ $Y_{p}^{\;r}$ represents the p^th^ feature set of the r^th^ layer. $Y_{o}^{r-1}$ represents the o^th^ feature map of the r−1 layer. $a_{p}^{r}$ is the bias. and f( ) represents the nonlinear activation function.

Flatten layer: In the flatten layer, the pooled featured map generated in the last pooling layer is converted to a one-dimensional feature map, and this one-dimensional featured map is the input to the fully connected layer.

Fully connected layer: The fully connected layer combines the features transmitted in the previous layer to achieve accuracy in the classification problem.

Output layer: The input image in the form of a set of features that are finally passed over to the output layer after crossing over the convolution, pooling and fully connected layers. The output layer is the classification layer in the form of probability. In the case of binary classifiers, a logistic regression model is used in the output layer; however, for a multiclass classification problem, the softmax classifier is commonly used [38,39]. The softmax function is a normalized exponential function. In the input dataset, let the training dataset consist of n number of tag samples such that $\left\{\left(y^{\left(1\right)},z^{\left(1\right)}\right),\left(y^{\left(2\right)},z^{\left(2\right)}\right)\ldots \left(y^{\left(n\right)},z^{\left(n\right)}\right)\right\}$; here, $z^{\left(i\right)}\in \left\{1,2..k\right\}$. Let y be the input data given, then the probability of j for each category p(z=j|y) can be determined using the hypothesis function given in Equation (3).
(3)$$h_{o}\left(y^{\left(i\right)}\right)=\left[\begin{array}{c} p\left(z^{\left(i\right)}=\frac{1}{y^{\left(i\right)}};\theta \right)\\ p\left(z^{\left(i\right)}=\frac{2}{y^{\left(i\right)}};\theta \right)\\ p\left(z^{\left(i\right)}=\frac{k}{y^{\left(i\right)}};\theta \right) \end{array}\right]=\frac{1}{\sum_{j=1}^{k}e^{\theta_{j}^{T}y^{\left(i\right)}}}\left[\begin{array}{c} e^{\theta_{1}^{T}y^{\left(i\right)}}\\ e^{\theta_{2}^{T}y^{\left(i\right)}}\\ e^{\theta_{k}^{T}y^{\left(i\right)}} \end{array}\right]$$

Here, $\theta_{1,}\theta_{2,}\ldots \theta_{k}\epsilon R^{m+1}$ is the parameter and $\frac{1}{\sum_{j=1}^{k}e^{\theta_{j}^{T}y^{\left(i\right)}}}$ is used for normalization, and it ensures the probability of all classes is 1.

The loss function is given in Equation (4).
(4)$$Loss\left(\theta \right)=-\frac{1}{n}\left[\sum_{i=1}^{n}\sum_{j=1}^{n}1\left\{z^{i}=j\right\}\log\frac{e^{\theta_{j}^{T}y^{\left(i\right)}}}{\sum_{l=1}^{k}e^{\theta_{j}^{T}y^{\left(i\right)}}}\right]$$

Here, the function 1{ } is defined as

1{if the expression value is true }=1,

1{if the expression value is false }=0,

The value of j in the 1{ } is j={1,2,…k }. The gradient equation of the loss function is given in Equation (5).
(5)$$\nabla_{\theta_{j}}Loss\left(\theta \right)=-\frac{1}{n}\sum_{i=1}^{n}\left[y^{\left(i\right)}(1\left\{z^{\left(i\right)}=j\right\}-p(z^{\left(i\right)}=j|y^{\left(i\right)};\theta ))\;\right]\;\;$$

Equation (6) is used to update $\theta_{j}$.
(6)$$\theta_{j}=\theta_{j}-\alpha \nabla_{\theta_{j}}Loss\left(\theta \right)$$

The probability function, used to determine the input y, belongs to class j and is given in Equation (7).
(7)$$p\left(z^{\left(i\right)}=j|y^{\left(i\right)};\theta \right)=\frac{e^{\theta_{j}^{T}y^{\left(i\right)}}}{\sum_{l=1}^{k}e^{\theta_{j}^{T}y^{\left(i\right)}}}$$

In this research work, the above general architecture is used to design a customized CNN model for the classification of the histopathological images of oral cancer. The proposed CNN model is compared with a few state-of-the-art models, such as VGG16, VGG19, Alex net, ResNet50, ResNet101, Mobile Net and Inception Net.

### 3.1. Dataset Used

The present study was conducted for the classification of the oral database into binary classes, such as benign and malignant. The oral dataset for this study was collected from a repository of normal oral cavities and OSCC [40].The dataset consists of 1224 histopathological images of an oral cavity. The total images available in this dataset are of two different categories. In the first category, all the images are of 100×magnification. In the second category, all the images are of 400×magnification. There are 528 images in the first category, out of which 439 images are of oral cancer and 89 are of normal oral images. In the second category, all the images are of 400×magnification. Here, a total of 696 images is available, out of which 495 images are of the malignant or cancerous type, and the rest of the 201 images belong to the normal or benign type. The detailed distribution of images available in the dataset is given in Table 1. Considering both categories, there are 1224 total images, out of which there are 290 total normal oral images and 934 total cancerous images. Figure 1 represents the sample images of each category. In the current study, all 1224 images in the category are considered for the histopathological oral cancer image classification purpose.

### 3.2. Image Preprocessing:

In the current study, the images are collected from the specified repository [40], which we discuss in Section 3.1. The images are of different quality and dimensions. Some images are perfect, whereas some images are with a noisy background. Preprocessing of images is thus required to reduce the impurities and for noise removal. We have used Gaussian blur for the removal of noise from the images and it also smooths the edges. The Gaussian filter is a low pass filter, which reduces noise or high-frequency components from the images [41]. Figure 2 shows an original histopathological oral image and the resultant image after the Gaussian filter is applied to the same image. After the Gaussian filter is applied to all the images, in the next step, all the images are converted to a fixed size of 128×128 as the images are of diverse sizes. After the preprocessing of the images, the images are used for the image augmentation process.

### 3.3. Image Augmentation

The classification task of medical images is getting cutting edge accuracy through the CNN approach; however, there are a few challenges to consider. One of the biggest challenges in medical image classification is the inadequate amount of training and testing data. The deep learning models require a huge dataset to overcome the problem of overfitting and network generalization. Generally, a deep learning model performs better with the balanced and large dataset. In this study, we considered the histopathological oral images dataset, which consists of a total of 1224 images of two classes, namely cancerous and non-cancerous. The dataset is also imbalanced. Thus, image augmentation technique is used to generate balance and an adequate dataset. The augmentation technique is used to generate new dataset from the existing one, without losing any important feature of the image [42]. In this study, we used rotation with a range of 40, height and width shift with a range of 0.2, and zooming and shirring with a range of 0.2. Using all these techniques for data augmentation leads to the generation of 8199 images out of 1224 images. The detail of the images generated is given in Table 2. From Table 2 it is observed that the imbalanced dataset is balanced with a nearly equal number of OSCC and normal images after the augmentation process is carried out. After augmentation, 4119 images are generated in OSCC class, and 4080 images are generated for the normal class. This new dataset with 8199 images is used for the classification task in the proposed CNN model.

### 3.4. Data Partition

The total number of images generated from Table 2 is 8199, and all these images are split into training and testing dataset with a ratio of 75% and 25% based on the train test split strategy [43]. After the train and test split, the number of images in the training dataset is 6149 and it is 2050 in the testing dataset. While splitting, the dataset to train and the test set, along with an original image and all its augmented images, are kept in the same folder, i.e., either in the train or test set, which ensure the train and test set to be disjointed. The training dataset is used for the initial training of the model by initializing the weights and it is also used for fine tuning of hyperparameters to improve the accuracy of the model [44,45]. The hyperparameters are selected once the model is trained using the train dataset; after that, the test dataset is used to evaluate the predictive accuracy of the model [46]. In this work, we use the mentioned train test split ratio 75% and 25% of the total 8199 images in the proposed CNN model as well as on predefined models, namely VGG16, VGG19, Alexnet, ResNet50, ResNet101, Mobile Net and Inception Net.

## 4. Proposed Methodology

In the present study, the classification task of histopathological oral images is carried out using the proposed methodology given in Figure 3. The proposed methodology consists of six stages: (1) dataset collection, which is already discussed in Section 3.1; (2) preprocessing of images, which is discussed in Section 3.2; (3) data augmentation, which is discussed in Section 3.3; (4) data partition for the training and testing set, as discussed in Section 3.4; (5) classification of images using proposed CNN and for the comparison purpose various predefined state-of-the-art models like VGG16, VGG19, Alexnet, ResNet50, ResNet101, Mobile Net and Inception Net are also used for the classification task; and finally (6) performance analysis of classification result is carried out over the output of proposed model with all other models considered. The detailed structure of the proposed CNN model used in stage 5 is given in Section 4.1.

### 4.1. Proposed 10-Layer CNN Architecture

The classification of histopathological images of oral cancer problem can be solved by using the proposed CNN architecture shown in the figure. As discussed in Section 3, the proposed CNN model also consists of six basic layers: (i) input layer, (ii) Convolution layer, (iii) pooling layer, (iv) flattening layer, (v) fully connected layer and (vi) output layer. In the input layer, the histopathological oral images are taken as input images. These images are converted to matrix of pixels. The size of the input layer is 128×128×3 pixels. In the proposed model, a total of 10 layers is considered, specifically eight convolution layers, one dropout layer, one flatten layer and two fully connected layers. Figure 4 represents the architecture of the proposed 10-layer CNN model. In the proposed model, six pooling layers and six batch normalization layers are also used. In each layer of convolution, one set of the feature map is generated using Equation (1).

In the first convolution layer, 32 filters are used with kernel size of 3×3. The second convolution layer also uses 32 filters with a 3×3 kernel size. The 3rd and 4th convolution layers use 64 filters with a kernel size of 3×3. The 5th and 6th convolution layers use 128 filters with a 3×3 kernel size. The 7th and 8th convolution layers use 256 filters with the same 3×3 kernel size. All the eight convolution layers use rectified linear (ReLu) activation function. The output of one layer is given as input to the next layer. After the 1st convolution layer, a set of one max pooling layer followed by a batch normalization layer is used, and the 1st set of the reduced feature map is generated. Equation (2) is used for the max pooling calculation. After the 2nd convolution layer, again one set of a max pooling and batch normalization layer is used. After the 3rd convolution layer, only one max pooling layer is used. The 4th and 5th convolution layer are connected parallelly and, in between them, there is no other layer. After that one set of a max pooling and batch normalization layer is used. The 6th convolution layer is followed by only one max pooling layer. Similarly, only one batch normalization layer is used after the 7th convolution layer. The 8th convolution layer is followed by global average pooling and one batch normalization layer. After all the convolution layers, a dense layer is used with sigmoid activation function followed by a flatten layer.

The flatten layer is used to convert the resultant output from the previous layer to a single dimension. After the flatten layer one dropout layer is used with a 30% dropout mechanism to overcome the problem of overfitting. Finally, in the output layer, one dense layer is used with the softmax activation function. The proposed model is trained using the ‘Adam’ optimizer and categorical cross entropy loss function is used as the model is proposed for binary classification task. The mathematical expression for the categorial cross entropy loss function is given in Equation (4), which is used in this output layer of the proposed model. Table 3 shows the summary of parameters used for the proposed model in detail.

## 5. Experimental Studies

In this work for the binary classification of the histopathological image of oral cavity, the proposed methodology along with the 10-layer CNN model is implemented using python on google colab platform. The google colab is a freely available platform and it runs entirely on cloud [47]. The Keras library and Tensorflow framework are used to implement the proposed work. For the execution of the code, the graphics processing unit (GPU) is used. The GPU is also freely available on the google colab platform. For the comparison analysis, along with the customized 10-layer CNN model, seven different types of predefined models, such as VGG16, VGG19, Alexnet, ResNet50, ResNet101, Mobile Net and Inception Net, are also implemented using python on the google colab platform.

### Hyper Parameter Setting of the 10-Layer CNN

In this section experimentally the idle hyperparameter is decided for the proposed 10-layer CNN model by considering the validation accuracy as a measure. The proposed model is trained using 100 epochs in the google colab platform, and it took 2 h to complete the training process. The 10-layer CNN is executed several times with different hyperparameter settings, to find out the idle hyperparameters for the model. The performance of the proposed model under different hyperparameter settings is given in Table 4.

The proposed configurations given in the Table 4 and highlighted in bold letters gave best result with the highest accuracy 0.9782 on the test dataset. From Table 4, it is observed that the proposed CNN model gives its best result with eight convolution layers, max pooling through the kernel size 3×3 and by using the dropout rate 0.3.

## 6. Performance Evaluation and Result Analysis

The performance evaluation and results are discussed using three various aspects: (1) statistical measures of confusion metrics, which are used for the evaluation of cross validation accuracy; (2) performance measure graph, which is specifically used for accuracy, and the loss graph, which is used for result analysis; and finally (3) comparison with other models available in the literature is deliberated to prove the superiority of the proposed model.

### 6.1. Performance Evaluation Using Statistical Measure

The performance evaluation of the proposed model is done using five statistical measures such as precision, recall, specificity, F-measure and, most importantly, accuracy. The mathematical notation for precision is given in Equation (8); similarly, Equations (9)–(12) are used for the mathematical notation of recall or sensitivity, specificity, F-measure and accuracy, respectively.
(8)$$Precision=\frac{TP}{TP+FP}$$
(9)$$Recall\;Or\;Sensitivity=\frac{TP}{TP+FN}$$
(10)$$Specificity=\frac{TN}{TN+FP}$$
(11)$$F-measure=2\times \frac{Precision\times Recall}{Precision+Recall}$$
(12)$$Accuracy=\frac{TP+TN}{TP+TN+FP+FN}$$

Here, TP indicates true positive and TN indicates true negative; similarly, *FP* is false positive and FN is false negative. For the performance evaluation of the proposed 10-layer CNN model, the error rate is also calculated, which is the complementary to the accuracy measure of the model. Equation (13) is used for mathematical notation of the error rate. The error rate gives the misclassification measure [48].
(13)Error rate=1−Accuarcy

To prove the superiority of the proposed 10-layer CNN, all the performance matrices mentioned above are evaluated for the proposed model as well as for the comparative models, namely VGG16, VGG19, Alexnet, ResNet50, ResNet101, Mobile Net and Inception Net. Figure 5 shows the performance evaluation matrix of all the comparative models along with the proposed model. From the figure it is observed that ResNet101, ResNet50, Alexnet, Mobile Net and Inception Net performed better as compared to VGG16 and VGG19. However, the proposed CNN model outperforms all the considered models with accuracy of 0.97, F-measure of 0.97, recall of 0.98, precision of 0.97 and specificity of 0.97. 

The performance of the proposed model is also presented in terms of bar chart of error rate and accuracy. In Figure 6, the error rate and accuracy are shown for all the considered models. From this figure, it can be seen that the proposed model achieved the highest accuracy of 0.97 with lowest error rate of 0.03, whereas VGG19 achieved the lowest accuracy of 0.71 with the highest error rate of 0.29. The various performance measures of all the comparative models along with the proposed CNN model are listed out in Table 5.

From the comparative analysis of various performance measures, mentioned in Table 5, it can be justified that the proposed 10-layer CNN model outperforms the other comparative model with the highest accuracy of 0.9782. However, the performance of Alexnet, Resnet50, ResNet101, Mobile Net and Inception Net is promising with accuracy of 0.88, 0.91, 0.89, 0.93 and 0.92, respectively. VGG16 and VGG19 achieved the lowest performance with an accuracy 0.74 and 0.71, respectively.

### 6.2. Result Analysis Using Performance Measure Graph

In this section, the result analysis of the proposed model is done using a performance measure graph, specifically the accuracy and loss graph, generated during the training and validation process for all the comparative models. A model with minimum loss signifies the best result. The minimum loss indicates that the model learns from the training and validation phase with a lower error rate. On the other hand, the maximum accuracy value indicates optimal results for the model [49].

Figure 7 highlights the accuracy and loss graph of VGG16. From Figure 7, it can be observed that VGG16 showed an average validation accuracy of 0.74 and validation loss comes to a constant value of 0.63.

The accuracy and loss graph of VGG19 is given in Figure 8. In Figure 8, VGG19 showed an improved accuracy with the range of 0.65 to 0.74; however, the validation loss is constant at 0.62. The results of VGG16 and VGG19 confirmed that these two models are not suitable for the considered dataset. 

Similarly, Resnet50 shows an accuracy improvement within the range of 0.52 to 0.91 and inversely the validation loss decreases to 0.25. From this study, it is observed that the Alexnet and ResNet50 models show promising performance. The accuracy and loss graphs of Alexnet and ResNet50 are represented in Figure 9 and Figure 10, respectively. Alexnet shows the validation accuracy with a range from 0.65 to 0.88 within 100 epochs, and the validation loss also decreases from 2.31 to 0.31. 

The accuracy and loss graph of ResNet101 is given in Figure 11. From the figure, it can be observe that the validation accuracy reached 0.89, whereas the training accuracy reached up to 0.99. The gap between the training and validation accuracy is more as compared to the Rasnet50 and Alexnet. For an optimum model, the gap between training and validation accuracy should be minimum. 

The accuracy and loss graph of Inception Net is represented in Figure 12. From the graph, it can be observed that the model achieved a training accuracy of 0.99 and validation accuracy of 0.92. The figure shows the gap between training and validation accuracy is less as compared to Resnet101. 

The accuracy and loss graph of the Mobile Net model is given in Figure 13. From the figure, it can be observed that the model achieved a validation accuracy of 0.93 and training accuracy of 0.99. From this observation, it can be concluded that Mobile Net is a better model as compared to the other models discussed above.

However, the proposed 10-layer CNN model achieved the highest validation accuracy of 0.97 with a validation loss decrease from 1.5 to 0.06. Figure 14 shows the proposed 10-layer CNN model’s accuracy and loss graph. In our study, none of the models shows the overfitting problem, which occurs when the training and validation loss are decreasing, but after a certain point, suddenly the validation loss increases. Similarly, none of the models undergoes the problem of underfitting, which occurs when there is a decrease in the training loss until the last epoch is reached. The proposed CNN model can be considered as a good fit model for our dataset as the training and validation loss had a minimal difference, and there is also a smaller gap between training and validation accuracy. Thus, the proposed model is considered as a suitable model for the dataset used.

### 6.3. Comparative Analysis with Various Models Available in Literature

To show the competence of the proposed model, the performance in terms of accuracy of the model is compared with the existing models available in the literature. Table 6 deliberates the comparative analysis of the various methods available in the literature with the proposed model. For the comparison, we chose only the models where deep learning approaches are considered and the dataset used for the classification task is restricted to histopathological images of oral cavity. By analyzing Table 6, it can be perceived that the proposed model is comparable with the recent literature. The proposed model achieved an accuracy of 0.9782 with an improvement of 0.32 as compared to the accuracy of model proposed by Navarun et al. (2020) [29]. As revealed from the past literature, limited work has been done in the area of classification of histopathological images of oral cavity. In this scenario, our proposed model would also be an added advantage in the particular field due to its high performance.

## 7. Conclusions

There are numerous codes of behavior for the detection and diagnosis of oral cancer by doctors. The detailed screening of the histopathological biopsy image is one of the major components to understand the diseases and for a better treatment. For the qualitative evaluation of the biopsy image, a skilled pathologist is required who can minutely differentiate between a healthy cell and a cancerous cell from the histopathological biopsy image of an oral cell. This process of qualitative and minute evaluation of biopsy image by the pathologist is a time-consuming process that results in a delay in disease detection and, hence, there will be a delay in treatment. In this aspect, there is a need for automated detection of OSCC to ensure a quick and correct diagnosis. In this current work, the authors propose the use of a deep learning models for the automatic detection of oral cancer from the oral biopsy histopathological image. The proposed 10-layer CNN model outperforms with the highest accuracy of 97.82% as compared to other state-of-the-art models, namely VGG16, VGG19, Alexnet, ResNet50, ResNet101, Inception Net and Mobile Net. The performance of the 10-layer CNN is also compared and analyzed with some of the recent work presented in the literature, and it is found that its performance is strong compared with the recent work done. From the above analysis, it concluded that the proposed 10-layer CNN model can be used as an automated tool to identify oral cancer and can help doctors as a supportive measure for the identification and treatment planning of oral cancer. For a future perspective, the proposed model can be extended to detect distinct stages of oral cancer, which can help both the patient and doctor to defeat cancer.

## Figures and Tables

**Figure 1 ijerph-20-02131-f001:**
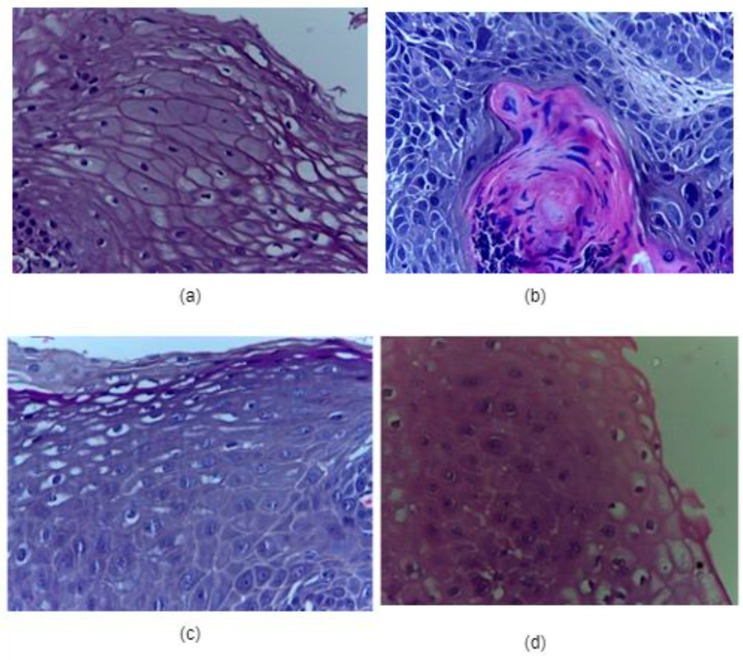
(**a**) Normal cell (100× magnification), (**b**) cancerous cell (100× magnification), (**c**) normal cell (400× magnification), (**d**) cancerous cell (400× magnification).

**Figure 2 ijerph-20-02131-f002:**
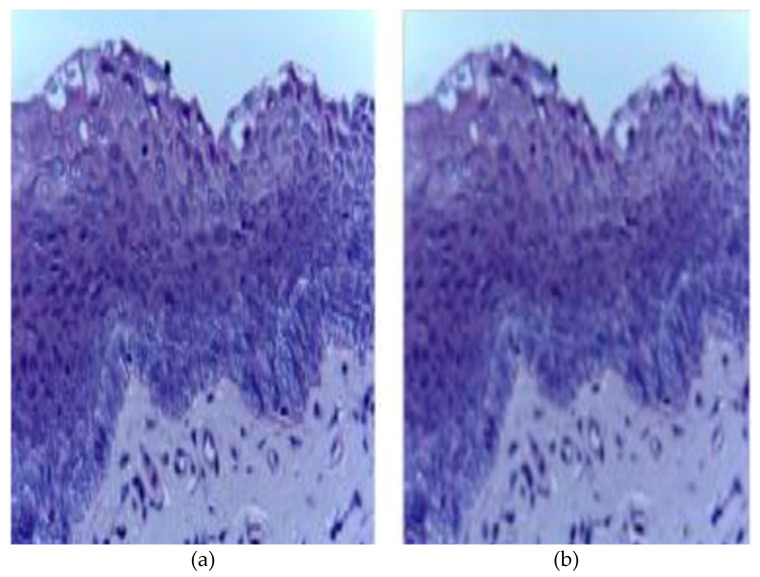
(**a**) Original histopathological oral image; (**b**) resultant image after Gaussian filter is applied.

**Figure 3 ijerph-20-02131-f003:**
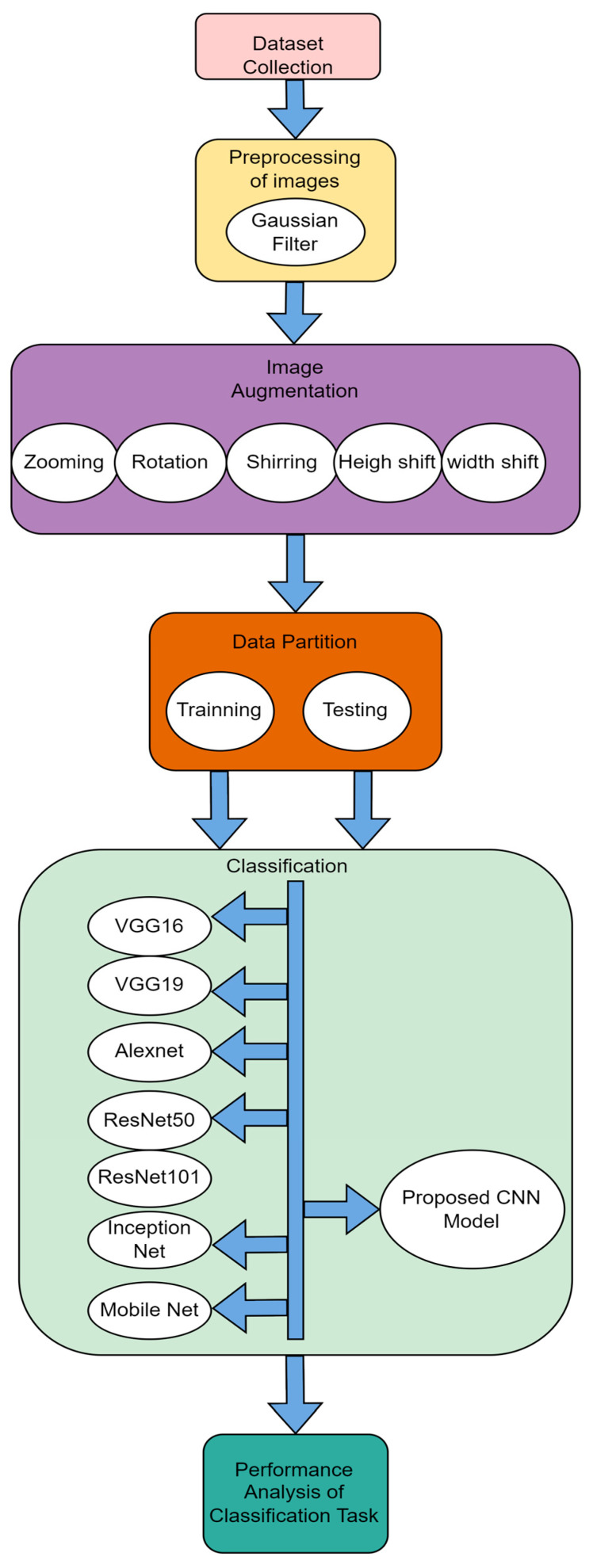
Proposed Methodology.

**Figure 4 ijerph-20-02131-f004:**
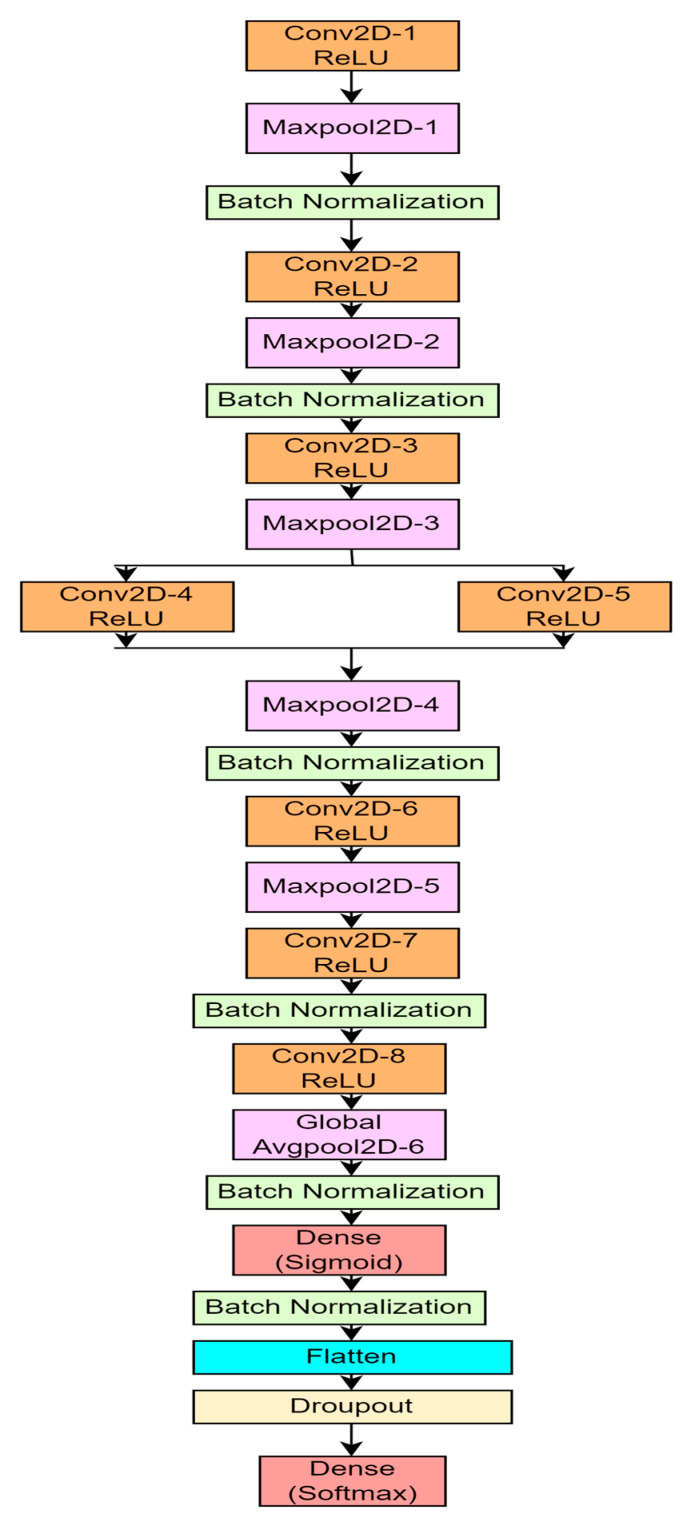
Architecture of proposed 10-layer CNN.

**Figure 5 ijerph-20-02131-f005:**
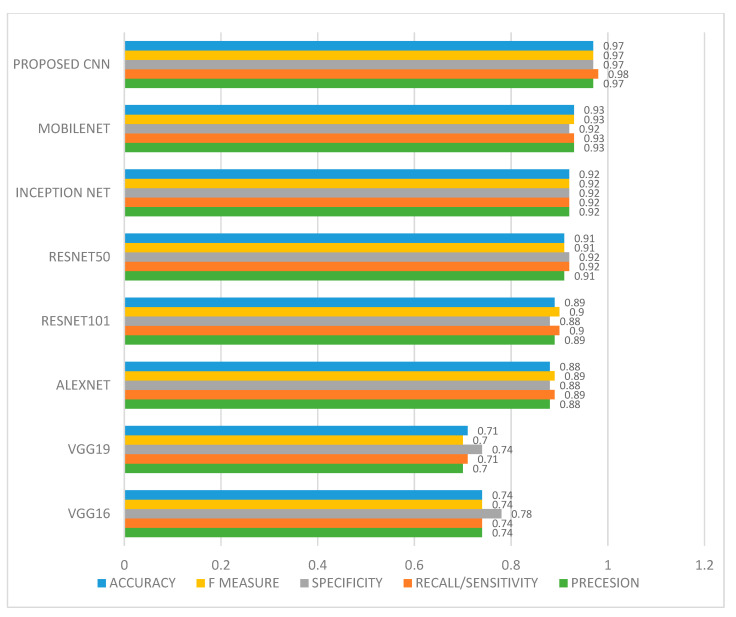
Performance evaluation using accuracy, F-measure, specificity, recall and precision.

**Figure 6 ijerph-20-02131-f006:**
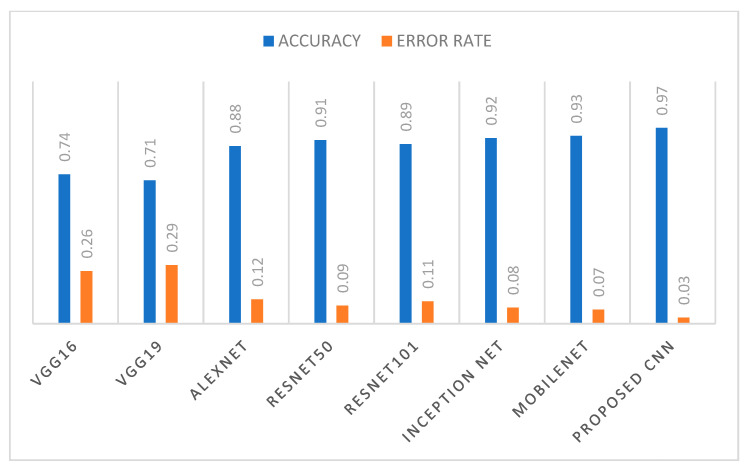
Comparison analysis of all the models using accuracy and error rate.

**Figure 7 ijerph-20-02131-f007:**
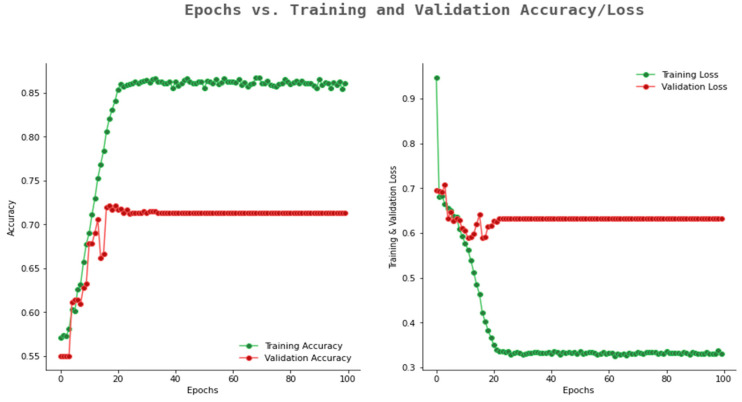
Accuracy and loss graph of VGG16.

**Figure 8 ijerph-20-02131-f008:**
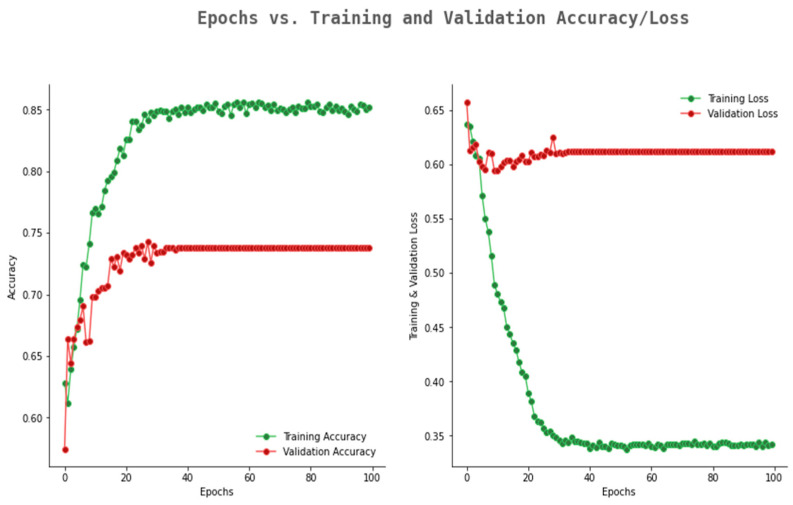
Accuracy loss graph of VGG19.

**Figure 9 ijerph-20-02131-f009:**
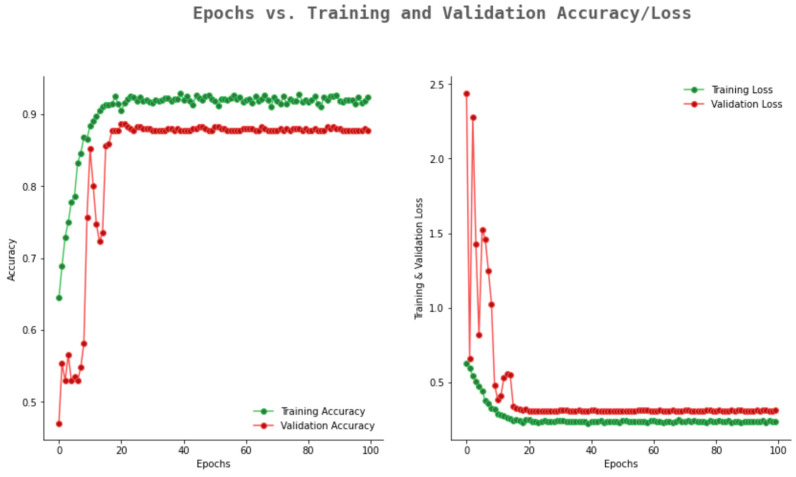
Accuracy and loss graph of Alexnet.

**Figure 10 ijerph-20-02131-f010:**
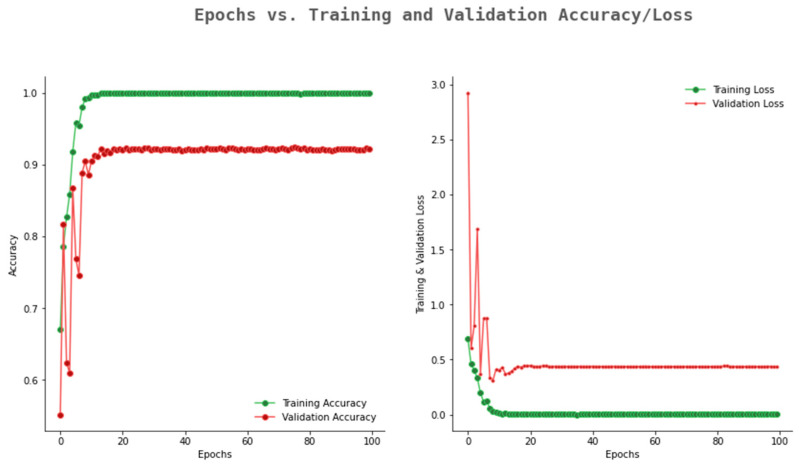
Accuracy and loss graph of ResNet50.

**Figure 11 ijerph-20-02131-f011:**
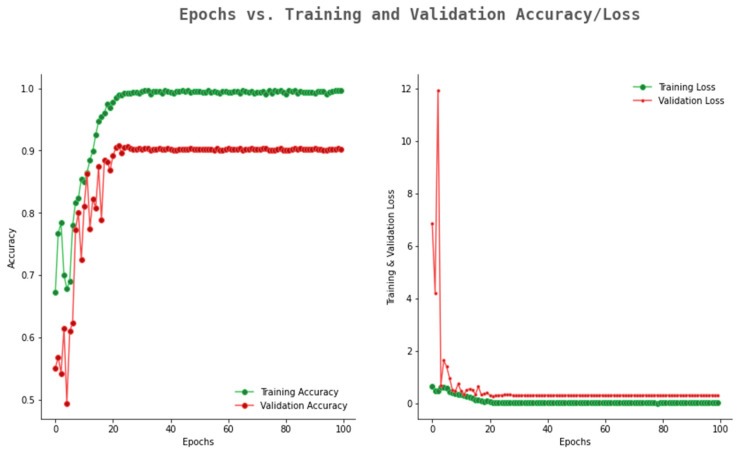
Accuracy and loss graph of ResNet101.

**Figure 12 ijerph-20-02131-f012:**
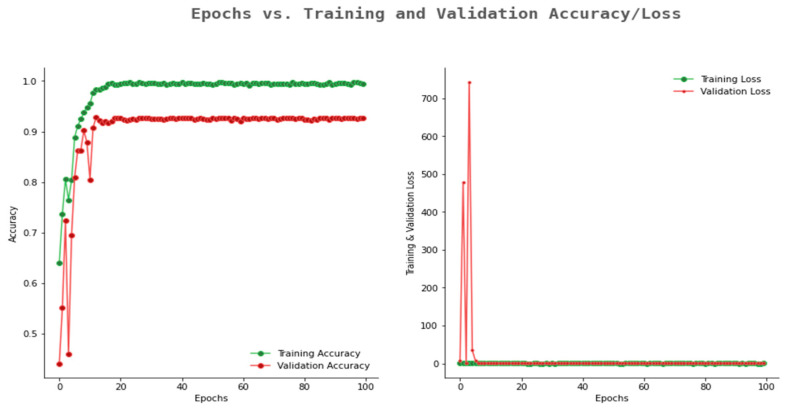
Accuracy and loss graph of Inception Net.

**Figure 13 ijerph-20-02131-f013:**
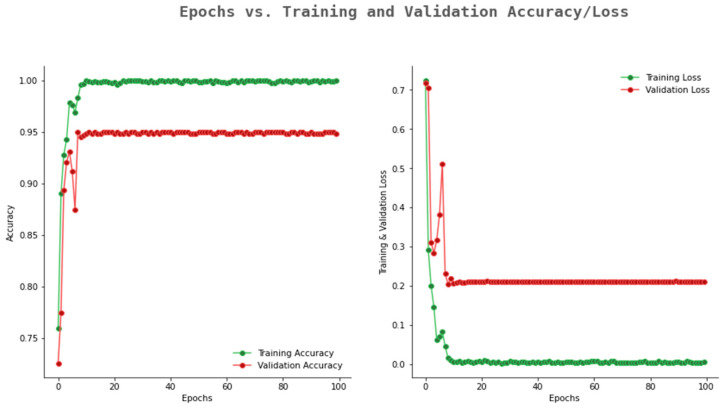
Accuracy and loss graph of Mobile Net.

**Figure 14 ijerph-20-02131-f014:**
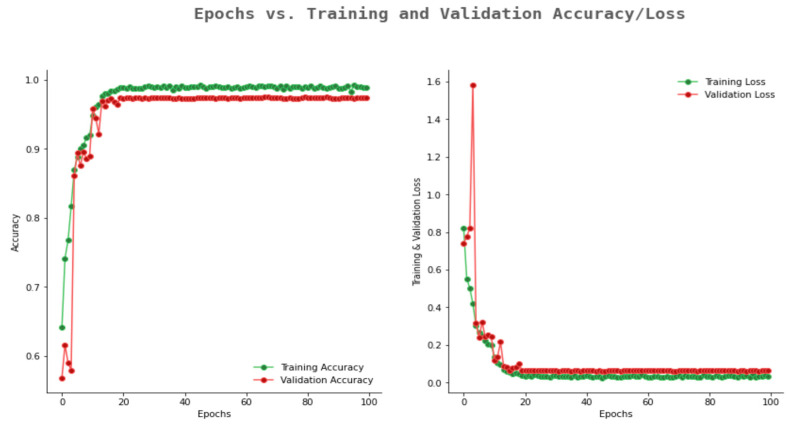
Accuracy and loss graph of the proposed 10-layer CNN model.

**Table 1 ijerph-20-02131-t001:** Distribution of images.

Category	Type	Total Quantity	Normal Images	Cancerous Images
Category-1	100× magnification	528	89	439
Category-2	400× magnification	696	201	495
Total images =		1224	290	934

**Table 2 ijerph-20-02131-t002:** Detail of image generation from augmentation process.

Category	No of Original OSCC Image	No. of New Images Generated from OSCC Images	No. of Original Non-Cancerous Image	No. of New Images Generated from Non-Cancerous Image
Category-1	439	2634	89	1869
Category-2	495	1485	201	2211
**Total**	934	4119	290	4080

**Table 3 ijerph-20-02131-t003:** Summary of parameter used in the proposed CNN.

Layer	Output Shape	Number of Kernel/Channel	Kernel Size	Stride	Padding	Parameter Generated
**Conv2D-1**	128×128	32	3×3	1	1	896
**Maxpooling2D-1**	64×64	32	3×3	2	1	0
**Batch Normalization**	64×64	32	3×3	1	1	128
**Conv2D-2**	64×64	32	3×3	1	1	9248
**Maxpooling2D-2**	32×32	32	3×3	2	1	0
**Batch Normalization**	32×32	32	3×3	1	1	128
**Conv2D-3**	32×32	64	3×3	1	1	18,496
**Maxpooling2D-3**		64	3×3	2	1	0
**Conv2D-4**	16×16	64	3×3	1	1	36,928
**Conv2D-5**	16×16	128	3×3	1	1	73,856
**Maxpooling2D-4**	8×8	128	3×3	2	1	0
**Batch Normalization**	8×8	128	3×3	1	1	512
**Conv2D-6**	8×8	128	3×3	1		147,584
**Maxpooling2D-5**	4×4	128	3×3	2	1	0
**Conv2D-7**	4×4	256	3×3	1	1	295,168
**Batch Normalization**	4×4	256	3×3	1	1	1024
**Conv2D-8**	4×4	256	3×3	1	1	590,080
**Global Average Pooling-6**	1×1	256	3×3	4	1	0
**Batch normalization**	1×1	256	3×3	1	1	1024
**Dense(sigmoid)**	1×1	1024	3×3	1	1	263,168
**Batch Normalization**	1×1	1024	3×3	1	1	4096
**Flatten**	1×1	1024	3×3	1	1	0
**Dropout**	1×1	1024	3×3	1	1	0
**Dense(softmax)**	1×1	2	3×3	1	1	2050
						Total=1,444,386

**Table 4 ijerph-20-02131-t004:** Performance of proposed CNN under different hyperparameters.

Number of Convolution Layers	Kernel Size	Pooling	Activation Function	Optimizer	Epoch	Dropout Rate	Training Accuracy	Validation Accuracy
5	5×5	Avg	ReLU	Adam	10	0.1	0.812	0.782
5	4×4	Avg	ReLU	Adam	50	0.2	0.812	0.772
5	3×3	Max	ReLU	Adam	100	0.3	0.851	0.813
6	5×5	Avg	ReLU	Adam	10	0.1	0.811	0.769
6	4×4	Avg	ReLU	Adam	50	0.2	0.817	0.771
6	3×3	Max	ReLU	Adam	100	0.3	0.857	0.825
7	5×5	Max	ReLU	Adam	10	0.1	0.887	0.850
7	4×4	Max	ReLU	Adam	50	0.2	0.935	0.892
7	3×3	Max	ReLU	Adam	100	0.3	0.982	0.942
8	5×5	Max	ReLU	Adam	10	0.1	0.992	0.955
8	4×4	Max	ReLU	Adam	50	0.2	0.992	0.969
**8**	3×3	**Max**	**ReLU**	**Adam**	**100**	**0.3**	**0.993**	**0.978**

**Table 5 ijerph-20-02131-t005:** Various performance measures of different models.

	VGG16	VGG19	ALEXNET	RESNET 50	RESNET 101	INCEPTION NET	MOBILENET	PROPOSED 10-LAYER CNN
**Precesion**	0.74	0.70	0.88	0.91	0.89	0.92	0.93	**0.97**
**Recall/** **Sensitivity**	0.74	0.71	0.89	0.92	0.90	0.92	0.93	**0.98**
**Specificity**	0.78	0.74	0.88	0.92	0.88	0.92	0.92	**0.97**
**F measure**	0.74	0.70	0.89	0.91	0.90	0.92	0.93	**0.97**
**AUC**	0.74	0.70	0.89	0.91	0.90	0.92	0.93	**0.97**
**Accuracy**	0.74	0.71	0.88	0.91	0.89	0.92	0.93	**0.97**
**Error rate**	0.26	0.29	0.12	0.09	0.11	0.08	0.07	**0.03**

**Table 6 ijerph-20-02131-t006:** Comparison of different existing models with the proposed model.

Existing Model in Literature	Method Used	Dataset Used	Accuracy in %
Navarun et al. (2020) [29]	CNN	Histopathological oral cavity images	97.50
Santisudha et al. (2019) [30]	CNN	Histopathological oral cavity images	96.77
Santisudha et al. (2020) [31]	Capsule Network	Histopathological oral cavity images	97.35
Proposed 10-layer CNN model	Customized CNN	Histopathological oral cavity images	**97.82**

## Data Availability

Not Applicable.

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
