# Peer review of "Automatic Detection of Oral Squamous Cell Carcinoma from Histopathological Images of Oral Mucosa Using Deep Convolutional Neural Network"

_ijerph, 2023, doi:10.3390/ijerph20032131_

Round 1
Reviewer 1 Report
The writing is very poor, many factual errors. Novelty is limited. Comparision with the existing work is not available. Just compare with other models. Not enough contribution. Must be improved.
Author Response
I am attaching the response to reviewer file.

Reviewer 2 Report
This manuscript by Das et al. developed a special CNN system to deal with the classification of Oral Squamous Cell Carcinoma. This is an interesting work, however, there are some problems with the description of your materials and methods. I recommend publication after the major revisions noted below.
1, In the manuscript, the authors mentioned a lot about the convolutional neural networks, and it costs about 6 pages to describe the CNN system by explaining the basic architectures of the CNN. Are there any differences between these architectures and the conventional CNN systems? If not, why this manuscript takes such a lot of space to talk about a conventional system? Only the creations need to be proposed here.
2, In the posted CNN system in Fig. 8 and Section 4.1, why do you call your new CNN system an 18-layer CNN system? The pooling layers and the normalization layers are commonly not included in the layer counting, so the name of your system should be changed. Another question is about the 4th and 5th convolutional layers, which is its main difference from Alexnet. Why do they seem to be parallel in the system? What is the advantage of this fixing? And, as even the ResNet was posted in 2015, which was 7 years ago, when talking about the related works, you may pay further attention to the newer systems like Mobilenet or ResNext, which might make your description more convincing.
3, Still about your new architecture. Commonly, ResNet performs better, as it takes the nested relationships between the outputs of the adjacent network layers, and takes steps to make sure that the outputs of the former layers are embedded into the inputs of the next residual block, so that the outputs will not departure from the original features and avoid the degeneration. However, the posted system did not consider it. So have you conducted any other tests to make sure that the system really performs better than ResNet, like taking another 100 epochs of training or conducting some further evaluations on more samples? Can you make a cross-validation on your dataset?
4, In Table 6, did you test all of these networks on the same dataset? If so, you may remind it in the experiments.
5, In Fig 12 and Fig 14, the original losses of the networks were too large so that the rest of the lines are invisible. Can you abandon such an extreme value, and make a more clear figure for the rest of the losses?
Author Response

(The authors gave the same response as above.)

Reviewer 3 Report
(1)This paper is well written in general. The author studied oral cancer using medical imaging and deep neural networks.The topic is important and readers in this field could benefit from the latest application of deep learning technologies.
(2)The authors further proposed a novel CNN architecture, for the specific OSCC image processing, the idea is relatively novel, the algorithm is effective.
Sample imbalance and feature analysis in disease prediction could be added to strengthen the literature review. Such as
"How to develop a more accurate risk prediction model when there are few events"
"Diabetes Mellitus risk prediction using age adaptation models"
(3)Popular CNN architectures were used and discussed in this paper. More recent advancement and approaches could be cited and further discussed, such as few-shot learning.
"Generalizing from a few examples: A survey on few-shot learning"
(4)The author has proposed a convolutional neural network model, for the automatic and early detection of OSCC.
This method is compared with several other models, the results seem promising. The intuition behind the design could be further discussed. Could the proposed model be generalized to other types of medical images? What is the difference between this OSCC image and images from general scenes?
Author Response

(The authors gave the same response as above.)

Reviewer 4 Report
The authors propose a method based on a CNN model to detect oral cancer from histopathological images. The proposed model is compared with other known models, and the reported results achieved with the proposed model outperform those obtained with other models.
Comments:
1. The wording should be thoroughly revised. There are several syntax and punctuation errors.
2. Figure 5 shows a clear difference between normal and cancerous images, suggesting that a human expert could perform this task without difficulty. How well does a human expert perform that task? The authors should explain to what extent the proposed model can help in clinical practice.
3. Line 36. Is it correct 12 million? Maybe it is 1.2 million?
4. Line 135. It is stated that "the results achieved from the above studies is not very promising". However, in [9] an accuracy of 96.3% is reported with the same dataset, which is very close to the results reported in the proposed method. Isn't that promising?
4. Section 3, lines 177 to 306. A detailed description of the fundamentals of convolutional networks is made. The "techniques used" section should focus on cutting-edge techniques and not on CNN's well-known fundamentals, which could simply be referenced. In my opinion, this section does not contribute anything to the manuscript.
5. Line 333. Reference [38] does not seem adequate if the goal is to make the reader understand the functioning of a Gaussian filter.
6. Sections 3.3 and 3.4. Data augmentation is performed using rotations, height and width shifts, zooming, and shirring. This process is applied to all the images, and after that, training/test splitting is performed.
First of all, the test set should be as close as possible to a real test; therefore, it should not contain synthetic images unless a special justification exists, in which case it should be explained.
Secondly (and this is my main concern regarding this study), synthetic images coming from the same real image should be in the same set (training or test). They should not be split, by any means, into different sets. Doing so incurs a severe methodological error since observations used in the training stage are also used to test the model, leading to optimistic results. Have the authors taken care of keeping all the images related to the same real image in the same dataset?
7. Line 362. Reference [40] does not seem adequate if the goal is to make the reader understand the functioning of the train/test split strategy.
8. Figure 11. It seems that the model is not learning anything. Is that a bug? It's strange. Same issue inn Fig. 12.
9. Pag. 21. Edit RasNet50 by ResNet50.
10. Figure 16. A ROC curve with only one operating point is depicted. A ROC curve with only one operating point is depicted. The AUC obtained in a curve of these characteristics is of little or null interest, given its lack of accuracy. This information can be better reported using specificity, sensitivity, and F-measure.
Author Response

(The authors gave the same response as above.)

Round 2
Reviewer 1 Report
The author's addressed most of my comments. I just want the native speaker editing to recommend it for publication.
Author Response
Thank you for the review and suggestions.
Reviewer 2 Report
I have no further question in this submission.
Author Response

(The authors gave the same response as above.)

Reviewer 4 Report
I appreciate the effort of the authors in considering or clarifying my comments. However, I still want to make sure about comment 10. I think there needs to be an error in how the ROC curve has been computed.
A ROC curve with a single operating point reveals that it has been constructed from data with only two possible values. ROC curves aim to show TPR vs. FPR at different thresholds; therefore, they are inadequate when built from data with just two possible values. Maybe the authors have used the predicted class instead of the probability of such prediction to construct the ROC curve? I'm just guessing, but in any case, it is not appropriate to use a ROC curve when the model output has only two possible values.
